# CNL–*Clitocybe nebularis* Lectin—The Fungal GalNAcβ1-4GlcNAc-Binding Lectin

**DOI:** 10.3390/molecules24234204

**Published:** 2019-11-20

**Authors:** Jerica Sabotič, Janko Kos

**Affiliations:** 1Department of Biotechnology, Jožef Stefan Institute, Jamova cesta 39, 1000 Ljubljana, Slovenia; janko.kos@ffa.uni-lj.si; 2Faculty of Pharmacy, University of Ljubljana, Aškerčeva cesta 7, 1000 Ljubljana, Slovenia

**Keywords:** beta-trefoil fold, LacDiNAc, entomotoxic, leukemic T-cell, antiproliferative, dendritic cell

## Abstract

*Clitocybe nebularis* lectin (CNL) is present in fruiting bodies of clouded agaric along with several similar isolectins that are all small and stable proteins. It is a beta-trefoil type lectin forming homodimers that are essential for its functionality. It binds specifically *N*,*N*′-diacetyllactosediamine (GalNAcβ1-4GlcNAc, LacDiNac) and human blood group A determinant-containing glycan epitopes. Its most probable function is to defend fruiting bodies against predators and parasites. In addition, an endogenous regulatory function is possible for CNL, as indicated by its interaction with fungal protease inhibitors sharing the beta-trefoil fold. CNL is toxic to insects, nematodes and amoebae, as well as to leukemic T-cell lines. Bivalent carbohydrate binding is essential for the toxicity of CNL, against both invertebrates and cancer-derived cell lines. In addition, CNL exhibits potent immunostimulation of human dendritic cells, resulting in a strong T helper cell type 1 response. Based on its unique characteristics, CNL is a promising candidate for applications in human and veterinary medicine as well as in agriculture, for plant protection.

## 1. Introduction

*Clitocybe nebularis* lectin (CNL) is one of many lectins identified in fruiting bodies of clouded agaric (*Clitocybe nebularis*) mushroom. They are small and stable proteins that display a variety of glycan binding specificities [1,2]. Obtaining the gene coding for CNL enabled production of the recombinant protein and, hence, detailed structural and functional analysis [3,4]. CNL is a hololectin that forms homodimers, which is essential for its functionality. Structural analysis revealed it to be a β-trefoil-type lectin. Glycan microarray analysis confirmed its specific binding of terminal GalNAc, preferentially in the context of *N*,*N*′-diacetyllactosediamine (GalNAcβ1-4GlcNAc, LacDiNac) and human blood group A determinant-containing glycan epitopes (Table 1), as well as weaker binding of other β-galactosides [3,4]. By the presence of specific target glycans on nematodes and cancer cells it was confirmed that CNL is nematotoxic and specifically cytotoxic for leukemic T-cells [4,5].

In this review an overview of current knowledge about CNL is provided, including a description of its structure, its mechanism of action, and the potential for its application.

## 2. CNL Structure

CNL is a β-trefoil-type lectin, as was predicted from the cDNA-deduced amino acid sequence (Figure 1) and based on the presence of the conserved QxW motif and the non-polar residues important for the hydrophobic core of the fold [3]. The β-trefoil fold is one of the most prevalent hololectin family folds present in higher fungi [6]. Twelve β-strands, forming three structurally similar repeats named α, β, and γ repeats, are related by pseudo-3-fold symmetry and form a six-stranded β-barrel (Figure 2). The signature QxW motif is conserved in the α and γ repeat, while it is absent from the β repeat of CNL and other similar fungal lectins (Figure 1). High sequence divergence is common for the β-trefoil fold as the major sequence requirement of the fold is the presence of 18 hydrophobic residues at distinct sites within the molecule that stabilize the barrel structure. The 12 β-strands are connected by loops differing greatly in sequence and length that constitute more than half of the molecule [4,7,8]. In CNL only the α repeat harbors a functional carbohydrate binding site, in which β2 and β3 strands form the bottom of the shallow binding pocket and are surrounded by β2-β3 and β3-β4 loops [4].

The fungal β-trefoil fold lectins show very high sequence variability (Figure 1). They overall share only 7% to 16% sequence identity (25% to 35% similarity) among the representatives that have been thoroughly characterized, including MpL (*Macrolepiota procera* lectin) [9], CCL2 (*Coprinopsis cinerea* lectin 2) [10], RSA (*Rhizoctonia solani* agglutinin) [11], SSA (*Sclerotinia sclerotiorum* agglutinin) [12], and BEL β-trefoil (*Boletus edulis* lectin β-trefoil) [13]. Lectins CNL, MpL, and RSA are exceptions as they share 23% to 26%identity (30% to 40% similarity). Interestingly, these proteins with the same 3D structure form dimers through different interfaces resulting in carbohydrate binding sites positioned on the opposite faces of the dimer (CNL and RSA) or on one face of the dimer (CCL2 and SSA), while BEL β-trefoil binds carbohydrates in all canonical binding sites [4,12,13,14,15].

Opposite the carbohydrate binding site lies the dimer interface area (Figure 3). The dimer is formed by interaction of the two β4-β5 loops through exclusively hydrophobic interactions and of the two β9-β10 loops through two hydrogen bonds. A double mutation was required to abolish dimerization, one that prevented the formation of hydrogen bonds (Asn110 to Asp) in β9-β10 loops and the other to disrupt the hydrophobic interactions (Leu54 to Arg or Trp) in β4-β5 loops. These monomeric mutants retained regular carbohydrate binding activity [4].

## 3. CNL Function

The biological function of CNL is probably to protect fruiting bodies against predators, grazers, and parasites. It has been shown to be toxic toward insects, nematodes, and amoebozoa [1,3,5].

### 3.1. CNL Function in Fruiting Body Defense

The small size of the protein, its solubility, and its resistance to high temperatures, extreme pH and proteolytic degradation are features associated with proteins known to be involved in fungal fruiting body defense [6]. Furthermore, several characteristics that CNL shares with other fungal fruiting body lectins indicate its cytoplasmic localization. Thus, no signal sequence for classical secretion is encoded in the CNL gene; CNL does not contain any cysteine residues and is not glycosylated. CNL is modified post-translationally by acetylation of its N-terminus [3,5,6]. In addition, it can interact with fungal protease inhibitors sharing the β-trefoil fold, modulating their activity. It is particularly interesting that CNL interacts with the trypsin-specific protease inhibitor cnispin, which is also expressed in fruiting bodies of *C. nebularis* [16,17]. Complex formation between CNL and cnispin has been demonstrated by glutaraldehyde cross-linking in vitro [17]. Moreover, cnispin inhibits the hemagglutination of human erythrocytes by CNL. Conversely, CNL did not prevent trypsin inhibition by cnispin in vitro, but did counter the activity of a similar trypsin-specific protease inhibitor, cospin, from *Coprinopsis cinerea* [17,18,19]. The interaction, with modulation of each other’s activity, suggests an important endogenous regulatory function, for both the CNL lectin and the protease inhibitors, that could also affect inter-organism interactions. Furthermore, the CNL and the protease inhibitors cnispin and cospin target non-endogenous glycan structures and proteases, respectively, indicating their roles in the defense of fruiting bodies [4,16,17,19].

### 3.2. The CNL Target Glycan

The CNL target glycan, LacDiNAc (Figure 4), is present abundantly in insects and parasitic helminths [20,21]. A negative effect on the development of insect larvae by CNL was therefore expected, and has been confirmed for the fruit fly (*Drosophila melanogaster*), mosquito (*Aedes aegypti*), and Colorado potato beetle (*Leptinotarsa decemlineata*) [1,5]. Furthermore, CNL showed moderate toxicity toward the amoebozoan *Acanthamoeba castellanii* and was toxic only to the *pmk*-1 mutant of the model nematode *Caenorhabditis elegans* that is hypersensitive toward a variety of stresses due to a defect in the p38 MAPK pathway [5].

The LacDiNAc terminal group is also found in *N*- and *O*-glycans of vertebrates, however it is rarely present in mammalian glycoproteins. It has been shown to play important roles in the regulation of half-life of the circulating glycohormone lutropin and in the functional differentiation of mammary epithelial cells. Furthermore, LacdiNAc on the non-reducing termini of *N*-glycans of cell surface glycoproteins significantly decreases or increases according to the progression stages of human cancers, making it a potentially useful biomarker for different types of cancer and their progression [22,23].

Only one other well characterized lectin that specifically binds the LacDiNac disaccharide has been described and this is *Wisteria floribunda* agglutinin (WFA) that belongs to leguminous lectin family and shares the fold with the well-known plant lectin ConA (Concanavalin A) that requires tightly bound calcium and manganese ions for interaction with carbohydrates [24,25].

## 4. Mechanisms of CNL Functions

### 4.1. Mechanism of Carbohydrate Binding

LacDiNAc binds to the carbohydrate binding pocket of the α repeat by six hydrogen bonds (Figure 5). Five are formed between four amino acid residues in CNL and two hydroxyl groups of galactose and the sixth between amino acid residue Ser24 in CNL and the N-acetyl group of GalNAc. Asp20, which is positioned at the bottom of the binding pocket, anchors the galactose by forming two hydrogen bonds with the O3 and O4 atoms of galactose. Further, the O3 atom forms two additional hydrogen bonds to amide nitrogen atoms in the side chains of residues Asn38 and Asn46, while the O4 atom forms a hydrogen bond with the amide nitrogen in Gly23. Additional water-mediated hydrogen bonds are also formed (Figure 5b). Lactose binds in the same way, through the five hydrogen bonds formed by galactose, however they are, in general, longer than those in structures of CNL in complex with LacDiNAc. The carbohydrate binding site is effectively inactivated by the single mutation of Asp20 to Arg, whose long side chain fills the binding pocket and prevents carbohydrate docking.

Glycan microarray analysis confirmed the higher affinity of CNL for LacDiNAc than for lactose. Additional quantitative binding data for these and other similar carbohydrates will be indispensable for further understanding of the binding affinity. Furthermore, recombinant CNL shows a more restricted specificity for LacDiNAc than the natural CNL isolated from *C. nebularis* fruiting bodies. This broader specificity is most probably the consequence of sequence heterogeneity in isolates of natural CNL that probably comprise multiple isolectins differing slightly in their carbohydrate binding specificities [3,4]. Limited sequence heterogeneity, resulting in considerable changes in function, was also shown for the fungal protease inhibitors with β-trefoil fold, mycocypins and mycospins [18,26,27,28].

The binding of LacDiNAc to WFA is similar to that of CNL as hydrogen bonds are formed exclusively to the GalNAc moiety of the disaccharide, however, seven hydrogen bonds are formed between WFA and GalNAc compared to the six in CNL. Similarly, indirect hydrogen bonds via water molecules and hydrophobic interactions are reported resulting in larger buried surface area of the GlcNAc moiety in LacDiNAc binding [25].

### 4.2. Mechanisms of Entomotoxicity, Nematotoxicity, and Leukemic T-Cell Toxicity

The entomotoxicity of CNL is dependent on carbohydrate binding, since the addition of lactose during toxicity tests abolished toxicity of CNL toward *D. melanogaster* larvae [1]. Using the non-carbohydrate-binding mutant CNL Asp20Arg, the indispensability of carbohydrate binding was confirmed for nematotoxicity also. However, carbohydrate binding is not sufficient and, using non-dimerizing CNL mutants, it was shown that CNL dimerization is equally important for nematotoxicity. The bivalent carbohydrate binding is, therefore, essential for CNL activity, suggesting that CNL toxicity is mediated by cross-linking of glycoligands thus, most probably, triggering a signaling pathway (Figure 6) [4].

The exact mechanism of toxicity has not yet been determined for CNL, however, studies of toxicity mechanisms of other fungal β-trefoil lectins indicate a similar mechanism for all. Binding of RSA to intestinal epithelial cells is necessary for entomotoxicity of RSA, where it is not internalized but intoxicated epithelial cells show symptoms of apoptosis, possibly caused by lectin-mediated activation of the respective signaling pathways [11,30,31]. Similarly, the entomotoxicity of SSA depends on its binding to insect midgut epithelial cells and triggering their death probably via a signal transduction pathway as it is not internalized [32]. In addition, glycan binding is essential for nematotoxicity of CCL2 and is similarly mediated by binding to nematode intestinal epithelium without being endocytosed and leading to disintegration of microvillar organization [6,10,33].

CNL exhibits an antiproliferative effect exclusively on leukemic T-cell lines, e.g., Jurkat, in a dose-dependent manner [3]. Similar to invertebrate toxicity, bivalent carbohydrate binding is essential for the cytotoxicity of CNL against leukemic T-cells (Figure 6). Preincubation of CNL with lactose, or the use of a non-carbohydrate-binding mutant of CNL, abolished the cytotoxic effect. Furthermore, using a non-dimerizing mutant in cytotoxicity analyses showed that only an active dimer can cause the toxic effect [4]. Several other lectins isolated from mushrooms have shown antiproliferative activity against different human cell lines [34]. Among the other mentioned β-trefoil fold lectins from mushrooms, RSA and SSA were shown to exhibit antiproliferative effect in cell lines, but it was only tested against insect-derived cell lines [30,32]. The antiproliferative effect of the BEL β-trefoil lectin exhibited different degrees against nine different human cancer cell lines. It most strongly inhibited the proliferation of the HepG-2 hepatocellular carcinoma cells followed by the CaCo-2 human colorectal adenocarcinoma and MCF-7 breast cancer cell lines. Carbohydrate binding and internalization into cells were shown to be important for the antiproliferative effect [13]. On the other hand, MpL was shown not to be toxic to eight different mammalian cell lines despite entering the cell via endocytosis. MpL also acted as a vehicle for internalization of associated proteins and accumulated in lysosomes [35].

### 4.3. Mechanism of Dendritic Cell Activation

CNL strongly stimulates maturation of dendritic cells (DCs) that are the dominant antigen-presenting cells in humans [36]. DCs are key regulators of both protective immune responses and tolerance to self-antigens. The maturation of DCs is a complex process characterized by the acquisition of a number of fundamental properties: Antigen processing and presentation, migration and T-cell co-stimulation. DCs display morphological, phenotypic, and functional variability, enabling their control of both immunity and tolerance, depending on their activation status and the microenvironment. Triggering of Toll-like receptors on DCs is thought to be critical for their functional maturation to immunogenic DCs and the priming of naïve T-cells in response to infection, and therefore coupling innate and adaptive immunity [37]. Stimulation of DC maturation by CNL is activated through the Toll-like receptor 4 (TLR4) pathway and induces activation of NF-κB and, also, p38 MAPK signaling. This results in mature DCs capable of generating a strong T helper type 1 (Th1)-polarized T-cell response. Th1 T-cells support and maintain cellular immune responses that effectively counter invading pathogens and cancer pathologies [36].

## 5. Conclusions

CNL possesses unique characteristics that make it a promising candidate for applications in medicine, biotechnology, and agriculture. Based on its distinct carbohydrate-binding specificity it could find a use in diagnostic and therapeutic applications in medicine, particularly for immunotherapy, and especially for specific targeting of leukemic T-cells and for treating T-cell mediated autoimmune and inflammatory disorders as well as hematopoietic malignancies. Further, in addition to its application in clinical diagnostics, CNL could be used in basic medical and biotechnological research, including analysis of malignant transformations and altered glycosylation patterns during tumor progression and metastasis, employing microarray analysis, lectin-based biosensors or affinity chromatography. Finally, based on its nematotoxicity and entomotoxicity, CNL could be used in plant protection against pathogens and pests or in veterinary and human medicine against parasites [2,6,38].

## Figures and Tables

**Figure 1 molecules-24-04204-f001:**
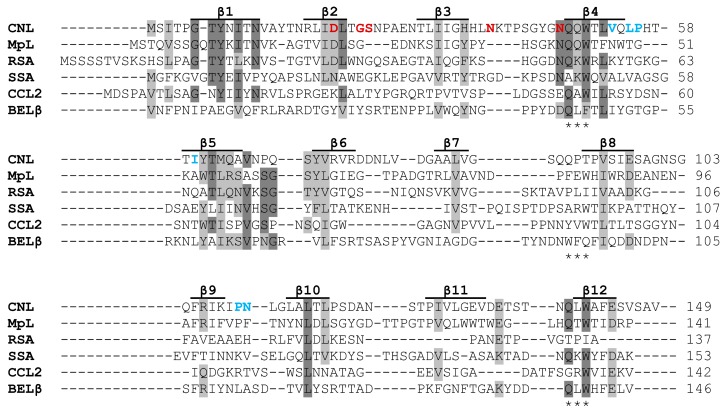
Amino acid sequence alignment of selected fungal β-trefoil fold lectins. Identical residues in at least four of the six sequences (60%) are highlighted in dark grey and similar residues are highlighted in light grey. The alignment was carried out using ClustalW with the BLOSUM62 matrix. The β-sheets of the secondary structure are indicated by lines above the sequences and the QxW motif is indicated by three asterisks below the sequence in each repeat. Amino acid residues involved in LacDiNAc binding are in red boldface and those involved in dimer formation are in blue.

**Figure 2 molecules-24-04204-f002:**
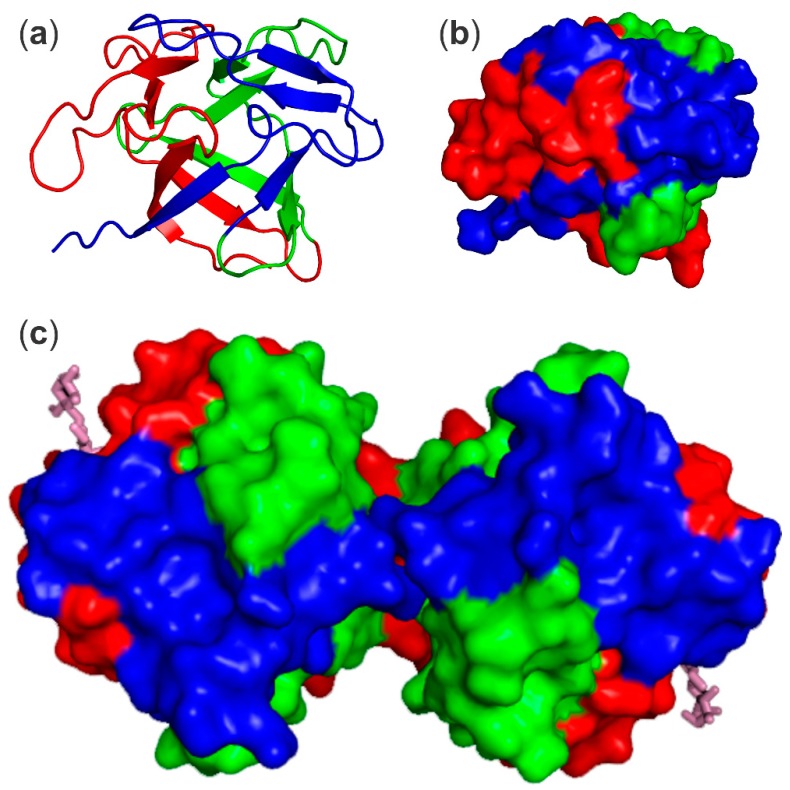
CNL three-dimensional (3D)-structure (PDB ID: 3NBD). (**a**) Structure of the CNL monomer that adopts the β-trefoil fold consisting of three repeats and shown in the following colors: α-repeat, red; β-repeat, green; γ-repeat, blue. (**b**) The solvent-accessible surface of CNL monomer in the same position as in (**a**). (**c**) The solvent-accessible surface area of CNL dimer showing monomers related by pseudo 2-fold rotational symmetry. The three repeats are colored as in (**a**). Lactose bound at the binding site on α-repeat is shown in magenta.

**Figure 3 molecules-24-04204-f003:**
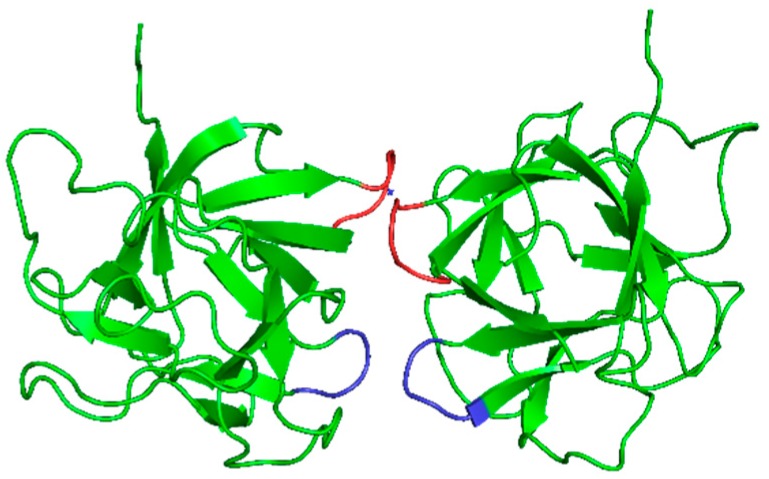
CNL dimer. The dimer interface comprises hydrophobic interactions between the β4 and β5 loops, shown in red, that involve residues Val52, Leu54, Pro55, and Ile59 and hydrogen bonds in the β9-β10 loops, shown in blue, that involve residues Pro109 and Asn110 from each loop.

**Figure 4 molecules-24-04204-f004:**
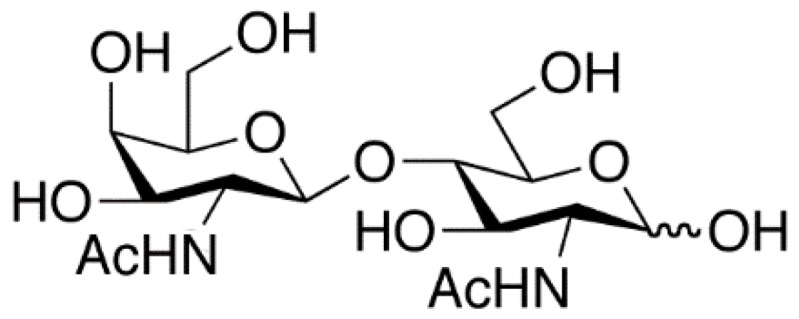
*N*,*N*′-diacetyllactosediamine (GalNAcβ1-4GlcNAc, LacDiNac).

**Figure 5 molecules-24-04204-f005:**
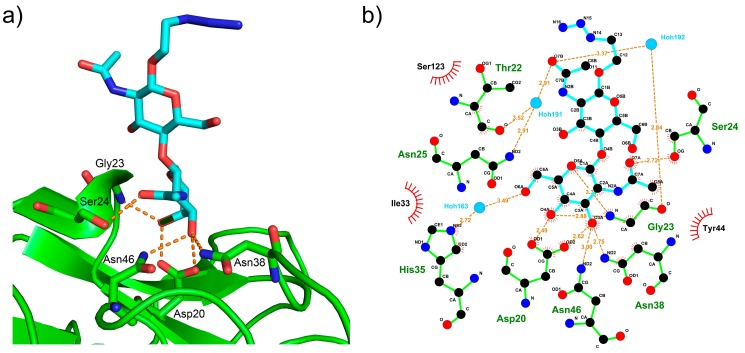
LacDiNAc bound to CNL (PDB ID 3NBE). Hydrogen bonds are shown as orange dashed lines and the amino acid residues involved in their formation are labeled. (**a**) 3D representation of the CNL carbohydrate-binding pocket. (**b**) A schematic representation of the CNL interaction with LacDiNAc generated by LIGPLOT [29]. Black, red and blue circles represent carbon, oxygen and nitrogen atoms, respectively, with green bonds in the amino acid residues and cyan bonds in the LacDiNAc ligand. Water molecules are represented by light blue circles. The spiked atoms are those involved in hydrophobic contacts and the spiked circle segments surround hydrophobic van der Waals partners.

**Figure 6 molecules-24-04204-f006:**
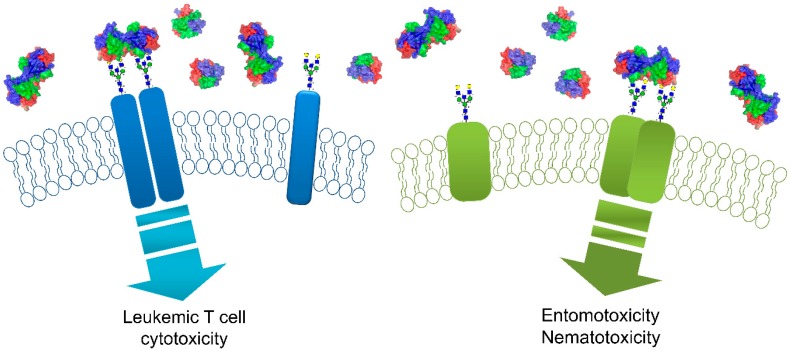
Proposed mechanism of CNL action. The bivalent carbohydrate-binding property of CNL, leading to the signal transduction pathway, is essential in both human leukemic T-cell specific cytotoxicity (left) and invertebrate toxicity (right).

**Table 1 molecules-24-04204-t001:** Glycan-binding specificity of *Clitocybe nebularis* lectin (CNL) determined by glycan microarray analysis. Specificity was defined using concentration-dependent binding of the lectin to glycan microarrays at different concentrations using mammalian printed array version 5.0 consisting of 611 glycan structures available at the Consortium for Functional Glycomics. The top 15 glycans showing highest affinities for CNL are shown [4]. Schematic representation of glycans was prepared by DrawGlycan-SNFG (http://www.virtualglycome.org/DrawGlycan/).

CNL Binding Glycans ^1^	Schematic Representation of Glycan Using the Symbolic Nomenclature for Glycans
GalNAcβ1-4GlcNAcβ1-3GalNAcβ1-4GlcNAcβ-sp ^2^	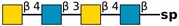
GalNAcβ1-4GlcNAcβ-sp	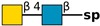
(6S)GalNAcβ1-4GlcNAcβ-sp	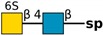
GalNAcβ1-4GlcNAcβ-sp	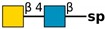
GalNAcβ1-4GlcNAcβ1-2Manα-sp	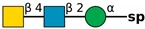
GalNAcβ1-4(6S)GlcNAcβ-sp	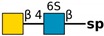
GalNAcβ1-3GlcNAcβ-sp	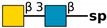
GalNAcα1-3[Fucα1-2]Galβ-sp	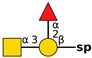
GalNAcα1-3[Fucα1-2]Galβ1-4GlcNAcβ1-2Manα1-6[GalNAcα1-3[Fucα1-2]Galβ1-4GlcNAcβ1-2Manα1-3]Manβ1-4GlcNAcβ1-4GlcNAcβ-sp	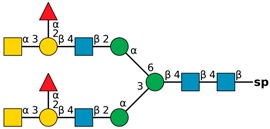
GalNAcα1-3[Fucα1-2]Galβ1-4GlcNAcβ-sp	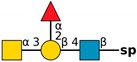
GalNAcβ-sp	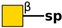
GalNAcα1-3[Fucα1-2]Galβ1-4GlcNAcβ1-2Manα-sp	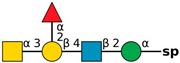
GalNAcα1-3[Fucα1-2]Galβ1-3GalNAcβ1-3Galα1-4Galβ1-4Glc-sp	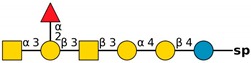
GalNAcα1-3[Fucα1-2]Galβ1-3GlcNAcβ1-2Manα1-6[GalNAcα1-3[Fucα1-2]Galβ1-3GlcNAcβ1-2Manα1-3]Manβ1-4GlcNAcβ1-4GlcNAcβ-sp	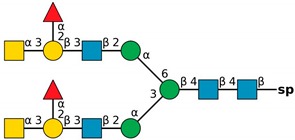
GalNAcα1-3[Fucα1-2]Galβ-sp	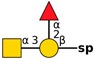

^1^ CNL binding to the Mammalian printed array version 5.0 at the Consortium for Functional Glycomics was assayed at 10, 1, and 0.1 μg/mL and a rank was determined for each glycan at each concentration. This list represents the top 15 glycans based on average ranks. ^2^ sp spacer.

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
