# Peer review of "CNL–Clitocybe nebularis Lectin—The Fungal GalNAcβ1-4GlcNAc-Binding Lectin"

_molecules, 2019, doi:10.3390/molecules24234204_

Round 1
Reviewer 1 Report
This is a very interesting paper, well written and easy to read. A minor comment: the PDB Id should be indicated in Figure 1 legend.
Author Response
Thank you for revieweing our manuscript and your kind words. We have included the PDB identifier in the Legend to Figure 1.
Reviewer 2 Report
The manuscript by Sabotic and Kos “CNL -Clitcybe nebularis lectin - the fungal …” is a well written review about this interesting fungal carbohydrate-binding protein. As there are many lectins with beneficial applications the field is very relevant. In contrast to plant lectins the investigation of fungal lectins is not that advanced yet, but it is gaining momentum and promises to find lectins with new specificities and new applications.
The manuscript is suitable for publishing in "molecules" after addressing the following points.
Major points:
1. The specificity of the lectin could be described clearer; the text and the title seem to deviate. Glycan array data from the references [3] and [4] reveal that the binding oligosaccharides almost exclusively have either a terminal GalNAc-alpha1,3-Gal (human blood group A) or GalNAc-beta1,4-GlcNAc (LacDiNac), but the title says the lectin is “GalNAc-beta1,4-GlcNAc-specific”, which is obviously not true since it excludes the human blood group A. The array data rather suggests that the lectin is just specific for one terminal GalNAc. Indeed the monosaccharide GalNAc, which is beta-linked to a spacer, is also binding quite well (Glycan #15 in reference [4]), supporting that CNL is only recognizing specifically a terminal GalNAc. In addition, the 3D structure of CNL in complex with LacDiNac (Fig. 5) shows that the protein contacts only the terminal GalNAc, suggesting that it does not matter to which saccharide the GalNAc is attached to nor if the linkage is in the alpha or beta configuration. For these reasons the title should rather state “… the fungal GalNAc-specific lectin”. The review might be improved by showing the consensus motifs of the glycan array schematically (with the typical symbols as found in Essentials of Glycobiology; https://www.ncbi.nlm.nih.gov/glycans/snfg.html)
2. Writing that the beta-trefoil fold is predicted from the cDNA-deduced amino acid, makes the beginning of this section a bit boring. If such predictions would be that good, why would we need then techniques like X-ray crystallography to determine the 3D structure? Why not just describing the characteristics of the beta-trefoil fold with the three QxW motifs and the residues in the core. In this context I suggest to cite the very detailed investigation of this fold by Murzin et al. 1992, J. Mol. Biol. 223, 531. Is it really true that from the CNL primary sequence the beta-trefoil fold can be predicted? Whereas the first and third QxW motifs are easily found in the CNL sequence, I have difficulties to find the second QxW motif. I also wonder whether the alignment in this region around residue 110 is correct, since the Trp residues of the other proteins do not align.
3. Discussing specific mutations of the binding site at the beginning of the manuscript and before showing the details of the interaction site is hard to understand. I suggest to rather mention mutations at the beginning rather unspecifically and discuss the details later after the 3D structure of the binding site is discussed. Alternatively, the 3D structure of the interaction site could be discussed earlier.
4. Are there any data for CNL available that mention the binding affinity like Kd, kON and kOFF rates? If yes, then they should be mentioned and discussed. If no, then this could be mentioned at the end in a kind of outlook suggesting experiments for the future. Quantitative binding data are quite important to understand the function of proteins in their biological context.
5. Alignment in Figure 3: it would be helpful for the reader if the important amino acids of the binding site and of the dimerization site would be highlighted, for example with colored letters in addition to the conservation in grey (at least for CNL). It would also be beneficial to see the secondary structure of CNL on top or below the alignment.
6. The text describing the interaction of CNL with protease inhibitors is quite confusing. It is mentioned that CNL interacts with cnipsin, but it is neither explained how this was detected nor which effect this causes. Only later in the paragraph cross-linking is mentioned, but it is not clear in which reference it is described (I guess [16]). The fact that the binding of CNL to the protease inhibitor does not show any effect in vitro on trypsin inhibition is even more confusing. Making the text clearer would help to get the message across.
7. Figure 5 should be mentioned earlier in section 1.4 (line 123), e.g. in the first sentence, because the description of the details will only be understood with this image. GalNAc does not have an “acetate” group (line 126) but rather an “N-acetyl” group. Figure 5 is quite important to understand the interactions; the amino acid side chains are nicely visible. However, the atoms of the sugar are hard to identify in this view. I suggest to present next to the 3D picture a schematic illustration of the interaction, for example generated by LIGPLOT or by hand. There would be space enough for two illustrations side by side.
Minor points:
-line 83-85: a reference for this sentence would be useful
-line 113: if it is a specific "glycohormone", I suggest to mention the name
-line 151: “carbohydrate-binding mutant”, mentioning the exact mutation would help
- sometimes a “the” seems to be missing, for example in
line 78: before “BLOSUM62 matrix”
line 96: before “CNL lectin” (because this is a specific lectin) and before “protease inhibitors” (because these are specific inhibitors)
line 174: where I would suggest to write “of the b-trefoil lectin BEL”
- sometimes the full journal names are given in the References, the correct abbreviations should be generally used
- There is one sugar in the glycan array data, which does not contain GalNAc, namely 3-sulfo-Gal-beta1,3-GlcNAc. That might be something interesting, maybe involving another binding site? However, it would need further investigation and since it is only a singular experimental result, it is fine not to mention it in the review.
Author Response
We are grateful for the very careful effort that has been made in the reviewing of our manuscript entitled “CNL -Clitcybe nebularis lectin - the fungal …”. We have followed reviewer’s suggestions and altered the manuscript accordingly. All the changes made are marked as track changes and responses to reviewer’s comments are given below. We hope that the changes and additional data included in the manuscript will satisfy the expectations and that the manuscript will be accepted for publication in Molecules.
Ad 1. As suggested by the reviewer, the title of the review was modified with regard to CNL specificity to read “…GalNAc-beta1,4-GlcNAc-binding lectin” to avoid ambiguity. Both groups of terminal glycans (human blood group A and LacDiNAc) were included in the abstract so there is no ambiguity there. Based on the top seven glycans bound on the microarray, which were all terminal GalNAc-beta1,4-GlcNAc (LacDiNAc), we decided against changing the title to “…GalNAc-specific lectin” but we did modify the Introduction section accordingly. As suggested by the reviewer we included a table of these highest binding glycans from the glycan microarray using the typical symbols.
Ad 2. Initially the fold of CNL was predicted from the amino acid sequence, which was later confirmed by X-ray crystallography. The reviewer was correct to find that the QxW motif is missing from the beta repeat and to make it clearer this information was added in the text along the beta-trefoil fold description. The alignment in the beta repeat cannot be much improved as the sequences are too heterogeneous.
Ad 3. The mentioning of specific mutation of the binding site was moved to the site where the 3D structure of the binding site is discussed as suggested by the reviewer.
Ad 4. We agree with the reviewer and a comment was added to the Mechanism of carbohydrate binding section.
Ad 5. The alignment was improved as suggested by the reviewer.
Ad 6. The text was modified to make it clearer.
Ad 7. The Figure 5 comments have been considered and the text altered accordingly with a LIGPLOT illustration included.
Ad Minor points. The minor points have all been addressed with the exception of changing the line 174 “of the b-trefoil lectin BEL”, as the name of the lectin is “BEL b-trefoil”.
Reviewer 3 Report
This is an excellent review work of Clitocybe Nebularis lectin regarding its structure, function, mechanism and potential applications.
Author Response
Thank you for reviewing our manuscript and your kind words.